# Informal caregivers and assistive technology in Norwegian nursing homes

**Camilla Anker-Hansen**[1]*, **Vigdis Abrahamsen Grøndahl**[1], **Ann Karin Helgesen**[1], **Liv Berit Fagerli**[1], **Guri Rummelhoff**[1], **Carina Bååth**[1,2], **Liv Halvorsrud**[1,3]

**1** Faculty of Health, Welfare and Organisation, Østfold University College, Halden, Norway, **2** Faculty of Health, Science and Technology, Department of Health Sciences, Karlstad University, Karlstad, Sweden, **3** Faculty of Health Science, Department of Nursing and Health Promotion, OsloMet, Oslo, Norway

☯ These authors contributed equally to this work.
* camilla.anker-hansen@hiof.no

## Abstract

### Aim

To explore informal caregivers' experiences and perspectives concerning assistive technology (AT) in two nursing homes, through the conceptual lens of person-centredness.

### Background

The integration and use of AT and a person-centred approach to care are political intentions within healthcare services, both internationally and in Norway. In nursing homes, informal caregivers are often collaborators with the staff, and can be important partners concerning the implementation of AT in a person-centred way. However, there is little knowledge about the informal caregivers' perspectives on the use of AT in nursing homes, or of whether or how they are included in the integration and use of AT.

### Methods

The study had a qualitative design and comprised eleven informal caregivers of residents in two nursing homes in Norway. In-depth interviews were used for data collection. The data were analysed using content analysis. COREQ reporting guidelines were applied to ensure comprehensive reporting.

### Results

Emerging themes highlighted the slow-going transition from old to new technology, and how the informal caregivers experienced that AT both promoted and degraded the dignity of their family members. Informal caregivers were positive to the use of technology, but have sparse knowledge and information about ATs in the nursing homes. They express a desire for AT to increase activity and safety, which promotes dignity, quality of life, and quality of the care for their family member. The informal caregivers want their family member to be seen, heard, and to get assistance on their own terms, even with regard to technology.

**Data Availability Statement:** Participants have not consented to have a transcription of their interview made publicly available, they only agreed that some extracts would be published. The authors will able to share some extracts only for academic

purposes. Data will be available upon request to Camilla Anker-Hansen, first author at [ca@hiof.no] or Kristian Sandbekk Norsted, Senior Advisor at ØUC, email: kristian.s.norsted@hiof.no. He is the Coordinator for the university's work with open research, and has no relationship to the data in this study.

**Funding:** The source of funding was financial support through internal fundings at Østfold University College (ØUC), and all the authors receive salary from the institution as we are all employees at ØUC. The funders had no role in study design, data collection and analysis, decision to publish, or preparation of the manuscript.

**Competing interests:** This study was financed by collaborational funding from Østfold University College. The funders had no role in study design, data collection and analysis, decision to publish, or preparation of the manuscript.The authors have declared that no competing interests exist. This does not alter our adherence to PLOS ONE policies on sharing data and materials

## Conclusion

Before AT can be implemented, informal caregivers need to be informed and listened to and included in the processes. Through their stories, one can form an idea of how important a person-centred approach is to contributing to individually tailored and introduced AT in collaboration with the informal caregivers.

## Introduction

Informal caregivers have an integral role in the quality of life of residents at nursing homes [1], and it is generally accepted that moving to a nursing home does not end the informal caregivers' involvement or their responsibilities [2]. Researchers have estimated that more than half of the children born after the turn of the millennium will reach the age of 100 years, and the number of people over the age of 67 is expected to have more than doubled by the year 2050 [3]. In line with age, there is also a significant increase in the chance of being affected by disease, including organic mental illness such as dementia. Dementia is an invasive disease that can have a pervasive effect not only on the person concerned and their immediate family, but also on the professionals caring for the affected person. Globally, more than 46 million people are reported to live with dementia [4]. A recent study found that over 100,000 Norwegians live with dementia, which is 25 per cent more than previously estimated [5]. Several people with dementia live in nursing homes, and it is estimated that about 80 per cent of those admitted to nursing homes have dementia [6].

The challenge for the health authorities is to develop health services that can handle the increasing need for health services, to increase the quality of care, to facilitate active ageing, and to ensure a good and dignified life for the person with dementia in collaboration with their informal caregivers [7]. Among the strategies proposed in government guidelines for healthcare services to solve these challenges are increased efficiency and greater use of technology (AT) [8,9]. Several terms are used in the literature and in healthcare policy to frame technology for supporting people in need of care such as, assistive technology [10], telecare [11], intelligent assistive technology [12], innovative assistive technology [13] and everyday technology [14]. The use of terms is challenging, but we use the internationally accepted term *assistive technology* (AT) in this study, which is in line with WHO's policy and can be defined as: "any item, piece of equipment, product or system that is used to increase, maintain or improve the functional capabilities and independence of people with cognitive, physical or communication difficulties" [10]. Furthermore, the use of ATs to inform and support active, safe and dignified lives is of importance for increasing the person's quality of life and the quality of healthcare [10,15]. The introduction of AT affects the people using it in both positive and negative ways, and research shows that new technology can be met with resistance from service providers, patients and informal caregivers [16].

The inclusion of informal caregivers can be an essential component in order to facilitate the introduction of AT for a person with dementia [17]. Enabling informal caregivers' involvement in AT, however, requires an organisation and care culture that is underpinned by person-centred values, where the informal caregivers are empowered to contribute according to individual wishes, and are seen as partners and not just a mean to achieve goals [18,19]. In many cases, the informal caregivers can provide important information about the patient's wishes, hopes and preferences, especially in cases where the patient is unable to express these on their own, both in general [20] and more specifically concerning the use of AT. Developing a fruitful collaboration between staff and informal caregivers can therefore be crucial regarding

the successful implementation and use of AT to improve the quality of care. A scoping review of informal caregivers' perspectives on person-centred care in nursing homes revealed short-comings in the quality of care, such as in communication with the staff and in the residents' quality of life [20]. The literature on both informal caregivers' and service users' perspectives on AT is growing [19,21–23], but to date there is a lack of knowledge regarding informal care-givers' experiences and perspectives explicitly concerning AT in nursing homes.

## Aim

The aim of this study is to explore informal caregivers' experiences and perspectives concern-ing AT in two nursing homes, through the conceptual lens of person-centredness.

## Methods

This study reports on the first stage of a two-year intervention project that focuses on improv-ing services and everyday life for residents in nursing homes and their informal caregivers, and facilitating the implementation of a person-centred approach in practice. The interviews in this study were conducted before the interventions started.

The study took a qualitative approach and was based on the consolidated criteria for report-ing qualitative research (COREQ) guidelines [24].

### Setting and participants

The setting was two nursing homes, located in the south-eastern part of Norway. In the first nurs-ing home, all wards were included in the study. The wards were all for people with dementia. In the second nursing home, the ward for people with dementia was included. The inclusion crite-rion for participation in the study was being an informal caregiver for a resident in one of the two included nursing homes. The recruitment of participants took place in two parallel processes, one for each nursing home. In the first nursing home, two of the authors (CAH & VAG) attended a gathering to which all patients and their families were invited. All those present were informed about the research project, and that we were seeking to recruit informal caregivers for interviews regarding their experiences as informal caregivers for residents in the nursing home. Further, they were informed that they could sign up for participation after the meeting, or later by giving notice to one of the head nurses. Nine persons signed up for participation on the same day, and one additional person signed up one month later. However, during the process of scheduling inter-views, three of these withdrew. One informal caregiver withdrew due to illness, another due to the loss of her mother, and the third person did not respond to inquiries. A further participant wanted her spouse to co-participate during the interview, and this request was granted.

In the second nursing home, the head nurse and the healthcare workers were informed about the project by three of the authors (CAH, VAG & LH) during an information meeting. Based on this information and an information letter, the head nurse recruited participants by asking all the patients' informal caregivers. All except one of the informal caregivers signed up for participation–a total of three persons.

All the informal caregivers who wanted to participate in the study were included, and the final sample consisted of eleven informal caregivers for ten patients in the nursing homes. Demographic characteristics of the participants are presented in Table 1.

### Data collection

The study comprised nine individual interviews, and one interview where the informant wanted her partner to participate in the interview with her. The interviewers suggested

**Table 1. Demographic characteristics.**

| Participants | | N = 11 |
|---|---|---|
| Age | Range | 35–78 (mean age = 58) |
| Sex | Male | 2 |
| | Female | 9 |
| Relationship to patient | Wife | 1 |
| | Daughter | 8 |
| | Son | 1 |
| | Other | 1 |

conducting the interviews in the nursing home, which all the informal caregivers accepted. An interview schedule was used during the interviews, which were conducted by two of the authors (CAH & LH). The interview schedule was prepared with detailed follow-up questions to ensure that comparable data were collected in all interviews (see supplement for interview schedule). Nevertheless, there was still flexibility to explore relevant matters that came up during the interviews. All interviews lasted approximately between 45–60 minutes, were digitally recorded, and were transcribed verbatim by an external transcriber, who had signed a non-disclosure agreement.

## Data analysis

Data were analysed by content analysis as described by Graneheim and Lundman [25]. All the interviews were read repeatedly to gain a sense of the whole picture. Text regarding informal caregivers' perspectives on AT was extracted from the transcriptions, which formed the unit of analysis. The extracts were then divided into meaning units, based on words, sentences or paragraphs that were connected by content and context [25]. Further, the meaning units were compressed into shorter sentences (condensed), although they remained close to the manifest content. Next, the condensed meaning units were coded. Codes with similar content were organised into two themes and four sub-themes, all representing the manifest content of the data. The analysis was performed by two of the authors (CAH & LH). CAH presented a draft of themes and sub-themes, which was discussed with LH. There was agreement between the authors on the topic that stood out, and the categories were fully developed in collaboration. Selected quotes are used in the presentation of the themes/sub-themes for illustration.

## Establishing trustworthiness

According to Lincoln and Guba [26], trustworthiness involves establishing four criteria: credibility, transferability, dependability and confirmability. To ensure trustworthiness and to prevent potential threats to validity, we used different strategies to establish each criterion as described by Lincoln and Guba [26]. Table 2 lists the strategies applied in this study.

## Preunderstandings

All members of the research group have a professional background in nursing, and together they have extensive research experience regarding informal caregivers, older people, AT and person-centredness. Five of the seven members have been working in nursing homes, but none of the research members have practical experience of the recently used AT in nursing homes at an operative level. Further, five members have been informal caregivers for a person in a nursing home. The professional and private backgrounds of the research group members contribute a comprehensive understanding of the general context of the study. However, the

**Table 2. Strategies applied to ensure trustworthiness.**

| Criterion | Strategy employed | Application in this study |
|---|---|---|
| Credibility | Triangulation | This study is a part of a two-year intervention project in two nursing homes. The perspectives of all involved in the care process have been considered in the project, and despite the fact that only relatives' perspectives appear in this study, the other acquired knowledge is also the basis for the researchers' understanding and approach to this study. For example, relatives' experiences were recognizable by what was said by other informants, as well as what the researchers themselves observed during their stay in the nursing homes. Thus, the triangulation was ensured through different types of informants and different sites. |
| Transferability | Providing thick description | In the study, emphasis is placed on providing background information that is necessary to understand the relevance and meanings that form the basis for relatives' experiences. |
| Dependability | A dense description of the research methodology | A detailed draft of the study protocol was made for the whole intervention project. In this study, it is emphasis on having rich description of the study methods. |
| Confirmability | Reflexivity<br>Audit trail | Regular meetings in the project group were held, where results were discussed and compared to other data in the project.<br>In the study protocol, description of the research process from the start of the project to the development and reporting of findings was drafted. In addition, condensed notes from all the data reported on in this study was made. |

established preunderstandings could also influence the analysis of the data material. With the objective of decreasing the risk of bias, awareness of this concern was raised in the group at an early stage, and possible preunderstandings and prejudices were debated ahead of the analysis process. This enabled a more conscious search for surprises in the material and to search for opinions and experiences that did not confirm the preunderstandings. such as the prejudice that relatives could be negative towards AT as this can be seen as a contribution to a care where the human contact is partly replaced by technology. However, what emerged in the analysis was that relatives were predominantly positive about AT, and even wanted more technology to be used.

Through personal reflections and discussions, the goal was to get past those preunderstandings, to be able to look for surprises in the material and to search for opinions and experiences that did not confirm the preunderstandings. One example where this was achieved was through the preunderstandings that male care partners are strong and demanding, and in general not satisfied with home care services. What was found in the material was that care partners silently accept what can be described as a poor service, and they are in general satisfied with home care services.

## Ethical considerations

Permission to conduct the study as a part of the project "Person-centred healthcare and technology—a complex intervention study in a nursing home in Norway" was provided by the heads of the two nursing homes included, and the further consent process was guided by the principles of the Helsinki Declaration (World Medical Association, 2013). The project was reported to the Norwegian Centre for Research Data (NSD) (Ref. no.291463), and approved by the Regional Committees for Medical and Health Research Ethics (REK 2019/41659). Information about the project was provided in oral and written form, and included participant anonymity and confidentiality, as well as the participants' right to withdraw from the study at any time until the data are included in the results. Written consent was collected from all the participants. The data material was treated in accordance with NSD's recommended procedures.

## Findings

This study aims to explore informal caregivers' experiences and perspectives concerning assistive technology in two nursing homes, through the conceptual lens of person-centredness. The findings are presented according to the two main themes, as illustrated in Table 3.

**Table 3. Themes and sub-themes.**

| Themes | Slow-going transition from old to new technology | AT promoting and degrading dignity |
|---|---|---|
| Sub-themes | • Discrepancy between status quo and desired progress<br>• Lack of knowledge and information | • Alienating AT<br>• Facilitating safety and activity |

## Slow-going transition from old to new technology

This theme illustrates the expression of the informal caregivers' that they thought it was a slow-going transition from old to new technology in the nursing home. This is illustrated through the two sub-themes: (i) "Discrepancy between status quo and desired progress", and (ii) "Lack of knowledge and information".

### Discrepancy between status quo and desired progress

One of the informal caregivers found it peculiar that basic technology, such as tablets and code locks on doors, were not in use in the nursing home.

> "A code lock would make give easier access to the patient ward." (Informal caregiver 4)

Others found it strange that AT that could promote the safety and activity level of the residents was not in place. One informal caregiver mentioned that her mother had had an activity bike in her previous nursing home. The bike could be linked to a monitor, which gave the illusion of cycling outdoors, for example through the streets of Paris. This had been a great way for her mother to exercise, but the device was not available in the nursing home where she now lived.

Another informal caregiver stated that her mother, who had dementia, had once disappeared from the nursing home and got lost. It took a long time to locate her and bring her safely back again, and, based on this experience, the daughter thought it would be useful to use a GPS tracking device. This is supported by another informal caregiver:

> "What I want, and what would make life easier for all of us, was that she had something that could locate her if she were to disappear from the nursing home." (Informal caregiver 9)

In addition, one informal caregiver focused on how the use of location technology could make it easy to track down missing residents, which could benefit both the residents and the staff:

> "Think about how it used to be earlier when people escaped -all the resources used to find them again! So, if you can get that on an app [location technology], the search will go much easier and faster, which is good for those who escape and those who are searching." (Informal caregiver 5)

One son talked about how the prevailing digital documentation platforms in the nursing home and in other health services did not correspond, which leads to undesirable situations. Once, his father had received the wrong medication in an ambulance transfer due to the paramedics not having access to information concerning his prescribed medication and allergies. He advocated for an e-system where all levels of the healthcare system can retrieve the required information. Further, he wanted the nursing home to use communication technology that could make it possible for him to get updates on the everyday life of his father. He mentioned regular phone calls and messages with pictures and/or videos as preferred solutions:

"It could be a system where no one else is photographed, but the staff take a picture of Dad, a video. . . nice weather and stuff like that, that would be great!" (Informal caregiver 1)

Another informal caregiver wanted to communicate with her father through a tablet, but was worried that this would be a burden on staff, because her father would need help to manage the device. She already had to constantly request updates regarding her father's health, and she felt that she was seen as a problem by the formal caregivers:

". . . I will not come with a tablet and say: 'help him call me' because I already feel I am a (. . .) a strain or a problem". (Informal caregiver 2)

A communication platform was also requested by this informal caregiver, who compared it with a platform used at her gym. The participant talked about how all the members of the group on the platform could get the same messages, give notice when they wanted to visit at the nursing home, etc., and therefore saw this as a useful communication tool. Further, she pointed out that this way of interacting has been used in other agencies for more than ten years.

## Lack of knowledge and information

AT was an unfamiliar term for most participants and many of them did not know what it was, what it entailed, or what it embraced. Nevertheless, they were all mostly positive towards the idea of it. They also had a lack of knowledge and information regarding the existing AT solutions in the nursing home:

"I do not know what they have here. I know there is an iPad upstairs [in the resident's room], I do not know what it does. What informal caregivers genuinely need is information, they [informal caregivers] are starving for information." (Informal caregiver 6)

"I: Let's talk about technology. Has there been anything, are you familiar with any technology [in the nursing home]?

R: No

I: Nothing. . .?

R: No." (Informal caregiver 3)

The above statements reflect a common experience among most of the participants. A recurring theme in most interviews was that the participants were not informed about what kind of AT was available in the nursing home, and they wanted more information. Only one informal caregiver said she was satisfied regarding the information about AT at the ward in which her mother lived. The lack of information was also apparent when it came to information regarding the family member of the informal caregiver:

"Suddenly the wristwatch came on–an alarm, probably a form of GPS. We did not receive any information–maybe we are not supposed to be informed? It was fine to make him use this watch, because if he runs away. . . (. . .). The watch is gone now, I do not know if he lost it or if they just stopped using it, but I thought, well, now he does not use the watch anymore." (Informal caregiver 8)

This quote illustrates how the informal caregiver was not informed when her father started to use an AT device, what it was exactly, or when and why he stopped using it.

## Technology promoting versus degrading dignity

This theme illustrates the informal caregivers' experience that the technology could both promote and degrade the dignity of their family members at the nursing home. This is illustrated through the two sub-themes: (i) "Alienating AT", and (ii) "Facilitating safety and activity".

### Alienating AT

One informal caregiver talked about the modern lighting system in the nursing home. She said her mother sat in the dark in her room because she did not know how to use the light switch. Another daughter also mentioned the lighting system as a challenge, because the automated light in the bathroom scared her mother:

> "There is a sensor for the light in the bathroom–when she opens the door, the light goes on in the bathroom, and she was completely terrified of that thing. It took months before she understood. . . . You can't make a 90-year-old lady familiar with new things when she's already a tad scared of the old things." (Informal caregiver 9)

Another informal caregiver talked about her own experience of fear regarding the fast development of AT in healthcare:

> "Regarding robots (. . .) and artificial intelligence. . . I'm a little scared of that, so if a robot should enter my husband's room and change his urinary catheter, no! But of course, you get used to that too, even if it seems strange now." (Informal caregiver 10)

She expressed a fear that robot technology would take over intimate care tasks her husband needed help with. However, she acknowledged that her concern could be related to this kind of AT being unfamiliar, and that her attitude could change over time.

There were also ATs in the nursing home that scared the residents or appeared as alienating devices, according to their informal caregivers. One example of this was the use of the remote control to operate the residents' personal TVs. Two informal caregivers explained how their family members were unable to watch TV because the remote control was unfamiliar, which made them afraid or incapable of using it. Another informal caregiver mentioned how the pre-installed AT in the resident's room made her mother afraid. Rails for a chairlift had been installed in the ceiling, from her bed to the bathroom. This device appeared frightening to her mother, and disturbed her sleep. In the end, the bed had to be moved to the other side of the room.

### Facilitating safety and activity

Several challenges regarding the use of AT have been identified so far, but the informal caregivers also saw clear benefits of using AT. They especially pointed out its potential contribution to an increasing degree of independence and physical activity for their family member. One daughter explained how her father, who was physically very fit, managed to climb over gates and fences of the nursing home area, and wandered off. Earlier, the residents at his ward could move freely in and out of the ward, as the outdoor area was an enclosed zone that was considered safe for them to be in unattended. However, after her father repeatedly managed to get out of the enclosed zone, the entrance to the ward was locked for everybody, and her father

was only allowed out with assistance. She welcomed localisation technology that could enable him to move around alone and safely, without being dependent on others to accompany him.

Another daughter highlighted how localisation technology could benefit both residents and healthcare professionals:

> "That would have been an advantage, a chip you put under the skin for example... Because I experience that, when she is as fit as she is, it is a minus for her in a way because she cannot enjoy life, they have to lock her inside because she can get out and get lost." (Informal caregiver 9)

She also felt that her mother could not enjoy life as she was locked up, because the staff were afraid she would get out of the nursing home and get lost. It was clear during the interview that this was a difficult matter for the daughter, and she said she would happily buy her mother tracking equipment herself to help the situation.

One informal caregiver described how her mother was kept locked up in her room to avoid unwanted visits from fellow residents–something that perhaps could have been remedied with the use of AT. The daughter asked for sensor technology that could trigger an alarm for the staff if someone entered a resident's room, in order to remedy the situation:

> "In the situation we have now, to have a technology where something [an alerting digital signal] is sent when someone enters her room, that would be much better than her being locked up." (Informal caregiver 6)

The daughter talked about how AT could help her mother avoid having to be locked in her room. She was also unsure whether her mother could pull the cord that was connected to the alarm system at the ward, and thought that a solution with sensors might rectify the situation.

Finally, one informal caregiver pointed out that a fall detection or tracking device for her father could be reassuring when he was moving around in the large areas of the nursing home. She pointed out that this could make movement both possible and safe for him:

> "It sounds interesting to use localisation technology, because he strolls, walk and fall a lot, he is dizzy and unsteady." (Informal caregiver 7)

## Discussion

The aim of this study was to explore informal caregivers' experiences and perspectives concerning assistive technology in two nursing homes, through the conceptual lens of person-centredness. From the analysis, the following two themes emerged: 1) Slow-going transition from old to new technology, and 2) AT promoting and degrading dignity. These two themes will be discussed below.

The findings show that the informal caregivers had scant knowledge of which ATs were used at the nursing homes, and they had received limited information from the staff. None of the informal caregivers mentioned the use of more recent ATs, such as location technology by sensor or GPS, digital medicine dispensers, or social robots for communication in the nursing home. Instead, they talked about well-known technology, such as telephones, lighting systems, and remote controls for televisions, with an emphasis on the challenges of this technology. Barriers to the implementation of technologies are reported in earlier studies [16,27]. One of the barriers identified is a lack of investment in the implementation of proper, inclusive training

of the frontline healthcare workers [27]. According to the informal caregivers' technological experiences in their own lives, some suggested the possible use of AT to improve the quality of care, such as improving the sharing of information both with the staff and with their family member. A recent scoping review highlighted the informal caregivers' experiences of short-comings in the information [20]. Sufficient information is crucial to obtaining information about the person with dementia, in order to stay involved in decision-making, and for advocacy for the family member. Furthermore, the informal caregivers are a heterogeneous group who need different types of information [19], which needs to be considered in the implementation of AT.

In research, as in clinical practice and in collaboration with informal caregivers, it is wise to distinguish between different types of AT and their potential. Gibson suggested a distinction between technology used *by*, *on* and *with* people with dementia [28]. Technology used *by* people with dementia were devices that the person used themselves in the early phases of dementia to cope with such daily tasks. Technologies used *on* persons with dementia were devices and systems used by informal or formal caregivers to care for a person with dementia; examples are monitoring systems, environmental sensors, cameras and alarms. Devices used *with* people with dementia involved a carer (either informal or formal caregiver), to enable social communication, entertainment or safety [28]. In our interviews, the focus of the informal caregivers soon turned to their family members and their interaction with AT when used *with* or *on*.

The concerns were mostly regarding how technology affected the everyday life of their family member at the nursing home. A point that is worth discussing is what the role of informal caregivers should be in this context. From the findings, we see that informal caregivers have received little training in the ATs (e.g. motion light sensors, computer tablets) that their family members had received at the nursing home. At the same time, it appears that informal caregivers make important observations with regard to both how ATs and everyday technology are experienced by the residents. This information could be important to communicate to the staff, especially in the cases where the residents were unable to do so on their own initiative. Using everyday technology, such as a remote control to operate the television was highlighted as a challenge by several informal caregivers. Some residents were unable to turn on the TV because the remote control was too complicated to use. Similar challenges have been identified in previous research, where devices, including remote controls in particular, are referred to as non-smart technology [29]. However, a recently published review claims that the individualisation process of the technology in dementia care research is growing, which indicates that research in this area is adopting a more co-creative and inclusive approach [30]. These authors also pointed out that sample sizes of the included studies were small, and there was a lack of standardised outcome measures. A future goal is to investigate this in larger samples, and to incorporate co-creative and inclusive approaches in clinical practice. The informal caregivers for persons with dementia might be included to a greater extent in the individual tailoring of ATs in nursing homes, and thereby be a resource for the implementation of AT in general. These assessments should be based on person-centred principles, where the individual's level of knowledge and interest in participation must be assessed in addition to the user of the AT's cognitive status.

When the user has a cognitive impairment, such as dementia in this particular study, it adds to the challenge of implementing new AT. Although the technology can be perceived as intimidating and complicated, it can also be difficult to master the new technology due to learning difficulties and failing memory [31]. New knowledge and skills can therefore be hard to integrate for a person with moderate or severe dementia. Quinn and Blandon [32] suggest that lifelong learning for people with dementia is disregarded due to cognitive decline, and needs to be focused on accordingly by using suitable teaching methods. With this point of

view and the possibilities of artificial technology [33] in mind, the use of AT in dementia care might change the everyday life of people with dementia, as well as of the informal caregivers in nursing homes.

However, it is important to recognise that AT can contribute to both the promotion and degradation of dignity, although there is scant focus on dignity in AT research [34]. In our interviews, matters of grave concern emerged. One informal caregiver said that her family member had been locked inside his room on several occasions, in order to be protected from a fellow resident. Another informal caregiver said that her mother was unable to leave her room without assistance or call for help due to physical impairment, but the door to her room was nevertheless also kept locked to keep fellow residents out. The Norwegian Regulations on Dignified Elderly Care [35] include the vague wording that the person "must have access to get out"–something that was clearly not provided in these examples. This raises a number of questions, such as the legality of such action, ethical aspects of the autonomy within the situation and the discretionary considerations that form the basis for such a practice, and the resources in healthcare. What is interesting in our study context, however, is that AT could have remedied the situation. Both informal caregivers in the example above expressed that AT represented desirable tools to prevent residents from being isolated inside their own rooms. It is relevant to question who the technology should be for. The localisation technology demanded by the informal caregivers exists, but instead of getting this in place, it was decided to contain residents by locking them inside their rooms. This may indicate that the technology is not necessarily used due to the needs of individual residents, but on the basis of other needs, such as efficiency and better working conditions for the employees. Nevertheless, it is important to keep in mind that people with dementia are sometimes unable to make autonomous decisions. A metasynthesis by Tranvåg et al. [36] describes that formal caregivers need to ensure that each person's essential needs are met and their dignity is protected. In the fulfilment of this, informal caregivers are crucial collaborators [20,37].

Findings also exemplify how technology was described as alienating by the informal caregivers, for example the lighting system. The light switches had a modern design that was unknown to one of the residents, which resulted in her not turning the light on in her room. Another resident was frightened by the automatic lighting system in the bathroom. AT should increase, maintain or improve the functional capabilities and independence of people with cognitive, physical or communication difficulties [15], but what if it instead adds to the user's everyday challenges? The examples above illustrate how some of the AT devices can contribute to insecurity for the residents and their informal caregivers, when perceived as something that creates insecurity and fear. Further, some informal caregivers discovered new AT installations that they had not been notified of in advance, that were supposed to create security but instead created more insecurity. This is in stark contrast to person-centred ideals, where a core value is that the person must come first [38]–a principle that embraces the patient, the informal caregiver and the staff [18].

Interestingly, none of the participants were concerned about privacy with regard to AT. On the contrary, some of them were willing to go quite far to ensure their loved one's safety, such as the informal caregiver who talked about chipping her mother so she could be monitored. This coincides with the findings of Sánchez et al. [22], where the need for welfare technology outweighed privacy concerns. The participants had no concerns about privacy regarding being monitored or other invasions of privacy, as long as it contributed to a sense of safety.

AT is often cited as the solution to tomorrow's problems of a shortage of healthcare workers, with regard to its efficiency [39]. A timely question, however, is whether technology always leads to increased efficiency. When people with a cognitive impairment are unable to turn their own television on or off, they require more assistance. When a nursing home resident

becomes scared and restless due to lights that inexplicably turn on and off, they require more attention from the staff. It is reasonable to assume that, almost no matter what devices the older nursing home residents with dementia are introduced to, this will trigger a need for assistance. The AT has no value in itself. It is through the way it is used, the competence of the person who uses it, and how it is experienced in use that is decisive for the utility value. A tablet lying in a drawer has no value to anyone. We need to solve this future labour shortage, but not by replacing care performed by humans with AT. Although, in many areas, society has developed technology that replaces people, it is emphasised from a political point of view that AT will not revolutionise the health services, meaning that technology will never be able to replace human care and physical proximity [9]. We suggest further development of AT in an ethically responsible matter, resolving both the effects and side-effects for the individual and society [40].

## Limitations

This study has some limitations that should be acknowledged. First, the small sample size (11 participants in 2 care homes) limits the findings to similar populations and contexts. Furthermore, some of the informal caregivers' perspectives were based solely on assumptions about AT, as they did not have any direct experience of it.

## Conclusion

This study set out to answer the question of identifying informal caregivers' experiences and perspectives concerning AT in nursing homes. The findings indicate that the informal caregivers have suggestions about the use of different types of AT and and their potential. Thus, they had little or no knowledge or information about the ATs that are either already in use or planned to be integrated in the nursing homes. The informal caregivers are positive to the use of both AT and everyday technology, and they express a wish for AT to increase activity and safety that promotes dignity, quality of life and quality of the care in a well-planned environment. They want the family member to be seen, heard, and to get help on their own terms, including when it comes to technology.

The informal caregivers' stories in this study represent the necessity of taking one step back before taking two steps forward. Before AT can be implemented, informal caregivers need to be informed and listened to, and included in the processes on behalf of themselves and, in some cases, of their family members. By referring to different AT experiences in their own lives, informal caregivers questioned why these ATs were not used in the nursing homes. Through their stories, one can form an idea of how important a person-centred approach is to individually tailor the AT in collaboration with the informal caregivers.

This study provides important knowledge about how to facilitate collaboration with informal caregivers in order to enhance good healthcare for residents of nursing homes. We believe that greater insight into informal caregivers' perceptions and expectations of AT will benefit healthcare recipients, providers and researchers, and stakeholders in the field of dementia.

## Acknowledgments

The authors wish to extend their sincere gratitude to all persons who participated in this study.

## Author Contributions

**Conceptualization:** Camilla Anker-Hansen, Vigdis Abrahamsen Grøndahl, Ann Karin Helgesen, Liv Berit Fagerli, Guri Rummelhoff, Carina Bååth, Liv Halvorsrud.

**Data curation:** Camilla Anker-Hansen.

**Formal analysis:** Camilla Anker-Hansen, Liv Halvorsrud.

**Methodology:** Camilla Anker-Hansen, Vigdis Abrahamsen Grøndahl, Ann Karin Helgesen, Liv Berit Fagerli, Guri Rummelhoff, Carina Bååth, Liv Halvorsrud.

**Writing – original draft:** Camilla Anker-Hansen, Liv Halvorsrud.

**Writing – review & editing:** Camilla Anker-Hansen, Vigdis Abrahamsen Grøndahl, Ann Karin Helgesen, Liv Berit Fagerli, Guri Rummelhoff, Carina Bååth, Liv Halvorsrud.

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
