## [Decision Letter · Decision Letter 0]

26 Apr 2022

PONE-D-22-07145Informal caregivers and assistive technology in Norwegian nursing homes.PLOS ONE

Dear Dr. Anker-Hansen,

Thank you for submitting your manuscript to PLOS ONE. After careful consideration, we feel that it has merit but does not fully meet PLOS ONE’s publication criteria as it currently stands. Therefore, we invite you to submit a revised version of the manuscript that addresses the points raised during the review process.

The article seems to propose an interesting topic, nevertheless the reported extracts from the interview with participants and the analysis provided should be enriched following the indication of the reviewers.

We look forward to receiving your revised manuscript.

Kind regards,

Simone Borsci, Ph.D.

Academic Editor

PLOS ONE

Journal Requirements:

"This study was financed by collaborational funding from Østfold University College."

"This study was financed by collaborational funding from Østfold University College. The funders had no role in study design, data collection and analysis, decision to publish, or preparation of the manuscript.The authors have declared that no competing interests exist."

6. We note that you have indicated that data from this study are available upon request. PLOS only allows data to be available upon request if there are legal or ethical restrictions on sharing data publicly. For more information on unacceptable data access restrictions, please see http://journals.plos.org/plosone/s/data-availability#loc-unacceptable-data-access-restrictions. 

7. Please ensure that you include a title page within your main document. You should list all authors and all affiliations as per our author instructions and clearly indicate the corresponding author.

8. Your ethics statement should only appear in the Methods section of your manuscript. If your ethics statement is written in any section besides the Methods, please move it to the Methods section and delete it from any other section. Please ensure that your ethics statement is included in your manuscript, as the ethics statement entered into the online submission form will not be published alongside your manuscript. 

Reviewers' comments:

Reviewer's Responses to Questions

**Comments to the Author**

1. Is the manuscript technically sound, and do the data support the conclusions?

Reviewer #1: Partly

Reviewer #2: Yes

2. Has the statistical analysis been performed appropriately and rigorously? 

Reviewer #1: N/A

Reviewer #2: Yes

3. Have the authors made all data underlying the findings in their manuscript fully available?

Reviewer #1: No

Reviewer #2: Yes

4. Is the manuscript presented in an intelligible fashion and written in standard English?

Reviewer #1: Yes

Reviewer #2: Yes

5. Review Comments to the Author

Reviewer #1: Thank you for the manuscript.

First of all I read the data availability statement regarding the authors not providing the full transcript of interviews and this seems reasonable from an ethical point of view, as since this was agreed with the participants. The many extracts relevant to the findings are included in the paper. However I would like the interview schedule to be included as supplementary material.

Major points:

The definition of AT on p.2 is broad and this is OK, but I think the paper could do more to provide a taxonomy, or at least a narrative summary, of the different ATs being talked about. Some of these are more obviously AT than others. Is a chip under the skin, smart lighting or remote controls considered an AT to most readers? They can be of course, by their application, but I think it would be helpful to say somewhere that some of the technologies discussed is conventional AT, some is smart home tech etc. A paper reviewing the types of AT used in dementia care could usefully be included in the citations and refer to the types of AT in the introduction.

The paper does not explain one very important thing. It is not explained in the paper if and how the interviewees were prompted about technologies. In the discussion the statement 'None of the informal caregivers mentioned the use of more recent ATs' suggested they were prompted as the extracts do show that the more recent ATs were mentioned. The interview schedule should be included and all of the prompts by the research team given.

Then, it should be made a lot clearer which extracts are spontaneous mentions of technologies and which were prompted by the researchers. If not it is hard to assess the level of knowledge of the interviewees and I have a concern that the caregivers were led too much by the researchers. In hindsight it might have been a good idea to have asked about different types of AT in a systematic way, to find out the level of knowledge about them.

The Conclusion should likewise be clearer about 'little or no knowledge' as it appears from the interviews that even if caregivers were not familiar with using more recent ATs they do have opinions about them. In the limitations, the statement that "perspectives were based solely on assumptions about AT" seems to me to be rather judgemental. From the extracts of the interviews, it appears that some caregivers did have direct experience of some technologies having been used in the nursing homes by their relatives (such as the example of the exercise bike, and the watch that was lost). So I am not sure it is fair to call this a limitation of the study. I think this assessment of knowledge would be better placed in the discussion and for the assessment not be so black and white about this point.

I think the paper should be revised to address the above points.

Minor points:

In the results, page 8, when the quote is given about caregiver 1 'she felt that she was seen as a problem', can it be clearer who they are problem to? I assume is it the nursing home staff but this is not specified.

The section headings and subheadings are explanatory but I would prefer to have some hierarchy so that METHODS and RESULTS are top level headings, with subheadings underneath. DISCUSSION is given in capitals as are CONCLUSIONS so this style could be followed earlier.

Typographical errors/suggestions for rewording:

Abstract:

Background - 'collaborators for the staff' -> 'collaborators with staff';

Results - 'Informal caregivers are positive to' -> 'Informal caregivers were positive about'

Introduction: para. 1: 'have a dementia disease' -> 'have dementia' or 'have a dementia diagnosis'; para 2: 'for the healthcare service' -> 'for healthcare services'; para. 3 'express this' -> 'express these'.

Data analysis: reference (22) is Graneheim, Lindgren, Lundman (or use et al.)

Establishing Trustworthiness: (23) does not need to be cited twice in the same paragraph.

Ethical considerations: NSD acronym needs to be expanded unless this is already in the redacted content.

Discussion: the point about sample sizes should be moved to limitations. The wording 'coercive clause' is unusual - can this be reworded. I think you mean 'coercive practice was used' and/or use the word 'containment'?

The sentence with 'no automaticity in ATs' is hard to follow and this could be reworded. I think you mean that using ATs does not remove the need for a person to be involved in the process of care.

Reviewer #2: The authors provide results from interviews about assistive technology (AT) with informal caregivers of loved ones with dementia living in 2 nursing homes in Norway. Authors find overall themes of slow-going transition from old technology to new technology and that AT can both promote and degrade dignity. They present illustrative quotes from the interviews to support more general statements. The manuscript will be strengthened if the authors consider the following points.

1. Authors should clarify why only one ward was included from the 2nd nursing home. Was this the only ward for people with dementia?

2. To give the readers a better sense of what was asked during the interviews, authors should provide the interview schedule and follow-up questions in supplemental material.

3. Table 2: not all readers will be familiar with the stated strategies. Authors should provide more detailed information about these strategies and their implementation in supplemental material.

4. In the Preunderstandings section, authors state that preunderstandings and prejudices were debated ahead of the analysis process. It is not clear what the result of this debate/discussion was and how that minimized bias in analyzing the interviews.

5. Authors include statements by 9 of the 11 informal caregivers. One is quoted 3 times. Authors may want to see if they can include quotes from each of the caregivers.

6. PLOS authors have the option to publish the peer review history of their article (what does this mean?). If published, this will include your full peer review and any attached files.

Reviewer #1: No

Reviewer #2: No

---

## [Author Response · Author response to Decision Letter 0]

15 Aug 2022

Word count 6244 without references.

Subject: PLOS ONE - Decision on Manuscript ID PONE-D-22-07145

Comments

 Answers

Comments to the Author 

1. Is the manuscript technically sound, and do the data support the conclusions?

The manuscript must describe a technically sound piece of scientific research with data that supports the conclusions. Experiments must have been conducted rigorously, with appropriate controls, replication, and sample sizes. The conclusions must be drawn appropriately based on the data presented. Reviewer #1: Partly

Reviewer #2: Yes #1 Thank you for reminding

us to strengthen the manuscript by carefully answering the editor and the reviewer’s questions and comments. 

3. Have the authors made all data underlying the findings in their manuscript fully available?

Reviewer #1: No

Reviewer #2: Yes #2 We fully agree that the data should be provided as part of the manuscript or its supporting information. However, this is few interviews in a small municipality in Norway. There are statements from the participants that can be identified. Due to ethical reasons, we do not add the datafile to this manuscript. 

See #19, #20

Is the manuscript presented in an intelligible fashion and written in standard English?

PLOS ONE does not copyedit accepted manuscripts, so the language in submitted articles must be clear, correct, and unambiguous. Any typographical or grammatical errors should be corrected at revision, so please note any specific errors here. Reviewer #1: Yes Reviewer #2: Yes #3 Thank you.

Reviewer 1 

First of all I read the data availability statement regarding the authors not providing the full transcript of interviews and this seems reasonable from an ethical point of view, as since this was agreed with the participants. The many extracts relevant to the findings are included in the paper. However I would like the interview schedule to be included as supplementary material. #4 Thank you for asking about the interview schedule. We have added this as supplementary material. See #2.

The definition of AT on p.2 is broad and this is OK, but I think the paper could do more to provide a taxonomy, or at least a narrative summary, of the different ATs being talked about. Some of these are more obviously AT than others. Is a chip under the skin, smart lighting or remote controls considered an AT to most readers? They can be of course, by their application, but I think it would be helpful to say somewhere that some of the technologies discussed is conventional AT, some is smart home tech etc. A paper reviewing the types of AT used in dementia care could usefully be included in the citations and refer to the types of AT in the introduction. #4 Thank you for valuable comments. 

Technologies to support older adults and citizens with disabilities can be divided into four domains: 1) for safety and security; 2) for coping with independent living; 3) health technologies; and 4) for well-being (1).

Several terms are used in the literature and in healthcare policy to frame technology for supporting people in need of care, such as, assistive technology (2, 3), assisted living technology (4), telecare (5), eHealth (6), intelligent assistive technology (7), innovative assistive technology (8), ambient assistive living (9) and everyday technology (10, 11). 

The translation of terms is challenging, but we use the internationally accepted term assistive technology in this study, which is in line with WHO’s policy. 

The term everyday technology was introduced in Sweden to cover household aids, electric tools, computers, TVs, mobile phones, ticket vending machines and self-service check-in at airports and hotels (10). 

In research, as in clinical practice, it is wise to distinguish between different types of assistive technology and their potential. Gibson (2016) suggested a distinction between technology used by, with and on people with dementia (12):

Such distinctions may be important for both researchers and for healthcare services when identifying suitable technology for assessed needs. However, it is not reasonable to expect that the informal caregiver kept this distinguish in mind when we asked about technology. 

We therefore clarify in the term technology in the introduction and in the discussion using Gibson (2016).

We have added more information about the technology on p 2, 13, 14 

The paper does not explain one very important thing. It is not explained in the paper if and how the interviewees were prompted about technologies. In the discussion the statement 'None of the informal caregivers mentioned the use of more recent ATs' suggested they were prompted as the extracts do show that the more recent ATs were mentioned. The interview schedule should be included and all of the prompts by the research team given.

Then, it should be made a lot clearer which extracts are spontaneous mentions of technologies and which were prompted by the researchers. If not it is hard to assess the level of knowledge of the interviewees and I have a concern that the caregivers were led too much by the researchers. In hindsight it might have been a good idea to have asked about different types of AT in a systematic way, to find out the level of knowledge about them.

 #5 These comments allow us to improve the manuscript. We have carefully considered others quotations about technology. We have added comments from interview no four. However, not from Informal caregiver no 11. He was together with his wife in the interview, and did not add anything of interest for this study. All the interviews are presented in then findings. See also #2 and #20. 

However, we asked broadly about technology and not asking about what was available at the nursing home. It was of importance to explore the informal caregivers` ideas about technology based on their knowledge and experiences. This approach may result in different types of technology being mentioned. See #4 

We have added the interview schedule, see #2 and #4

The Conclusion should likewise be clearer about 'little or no knowledge' as it appears from the interviews that even if caregivers were not familiar with using more recent ATs they do have opinions about them. In the limitations, the statement that "perspectives were based solely on assumptions about AT" seems to me to be rather judgemental. From the extracts of the interviews, it appears that some caregivers did have direct experience of some technologies having been used in the nursing homes by their relatives (such as the example of the exercise bike, and the watch that was lost). So I am not sure it is fair to call this a limitation of the study. I think this assessment of knowledge would be better placed in the discussion and for the assessment not be so black and white about this point. #6 Thank you, we have changed the conclusion. 

See p 17

In the results, page 8, when the quote is given about caregiver 1 'she felt that she was seen as a problem', can it be clearer who they are problem to? I assume is it the nursing home staff but this is not specified.

 #7

This is rewritten. 

The section headings and subheadings are explanatory but I would prefer to have some hierarchy so that METHODS and RESULTS are top level headings, with subheadings underneath. DISCUSSION is given in capitals as are CONCLUSIONS so this style could be followed earlier. #8 

All amended

Typographical errors/suggestions for rewording:

Abstract:

Background - 'collaborators for the staff' -> 'collaborators with staff';

Results - 'Informal caregivers are positive to' -> 'Informal caregivers were positive about' #9 All admitted

Introduction: para. 1: 'have a dementia disease' -> 'have dementia' or 'have a dementia diagnosis'; para 2: 'for the healthcare service' -> 'for healthcare services'; para. 3 'express this' -> 'express these'. #10 

1: 'have a dementia disease' is rewritten: 'have dementia'

2: rewritten

3: rewritten

Data analysis: reference (22) is Graneheim, Lindgren, Lundman (or use et al.) #11 

Rewritten: Graneheim et al.

Ethical considerations: NSD acronym needs to be expanded unless this is already in the redacted content. #12 This is already in the redacted content.

Discussion: the point about sample sizes should be moved to limitations?

 #13 Done, see p 17

The wording 'coercive clause' is unusual - can this be reworded. I think you mean 'coercive practice was used' and/or use the word 'containment'? #14 

'coercive clause' is rewritten ‘coercive practice was used', p 15

The sentence with 'no automaticity in ATs' is hard to follow and this could be reworded. I think you mean that using ATs does not remove the need for a person to be involved in the process of care #15 The sentence is omitted

Reviewer 2 

1. Authors should clarify why only one ward was included from the 2nd nursing home. Was this the only ward for people with dementia? #16 Done, see p 4.

2. To give the readers a better sense of what was asked during the interviews, authors should provide the interview schedule and follow-up questions in supplemental material. #17, See #2 and #4

3. Table 2: not all readers will be familiar with the stated strategies. Authors should provide more detailed information about these strategies and their implementation in supplemental material. #18, Amended, please see additional information in table 2. 

4. In the Preunderstandings section, authors state that preunderstandings and prejudices were debated ahead of the analysis process. It is not clear what the result of this debate/discussion was and how that minimized bias in analyzing the interviews. #19 – Amended, please see p. 6.

5. Authors include statements by 9 of the 11 informal caregivers. One is quoted 3 times. Authors may want to see if they can include quotes from each of the caregivers. #20 We have added information from all the interviews, p 7 and . See #2, and #5.

1. NOU 2011:11. Innovasjon i omsorg (Innovation in the Care Services). Ministry of Health and Care Services (Helse- og omsorgsdepartementet).

2. WHO. Assistive technology. 2018 [Available from: https://www.who.int/news-room/fact-sheets/detail/assistive-technology.

3. WFOT. Position Statement: Occupational Therapy and Assistive Technology 2019 [Available from: https://www.wfot.org/resources/occupational-therapy-and-assistive-technology.

4. Forsberg E-M, Thorstensen E, Casagrande FD, Holthe T, Halvorsrud LT, Lund A, et al. Is RRI a new R&I logic? A reflection from an integrated RRI project. 2020.

5. Berge MS. Telecare acceptance as sticky entrapment: A realist review. Gerontechnology. 2016;15(2):98-108.

6. Jakobsson E, Nygård L, Kottorp A, Malinowsky M. Experiences from using eHealth in contact with health care among older adults with cognitive impairment. Scand J Caring Sci. 2019;33:380-9.

7. Ienca M, Wangmo T, Jotterand F, Kressig RW, Elger B. Ethical Design of Intelligent Assistive Technologies for Dementia: A Descriptive Review. Science and engineering ethics. 2018;24(4):1035-55.

8. Thordardottir B, Malmgren Fänge A, Lethin C, Rodriguez Gatta D, Chiatti C. Acceptance and Use of Innovative Assistive Technologies among People with Cognitive Impairment and Their Caregivers: A Systematic Review. BioMed Research International. 2019;2019.

9. Nordic Innovation and Nordic Welfare Centre. Nordic ambient assistive living. Welfare technologies for active and independent living at home. 2019.

10. Nygård L, Rosenberg L, Kottorp A, inventorsEveryday Technology Use Questionnaire - ETUQ. Sweden2015 25.01.2016.

11. Malinowsky C, Kottorp A, Nygard L. Everyday technologies' levels of difficulty when used by older adults with and without cognitive impairment-Comparison of self-perceived versus observed difficulty estimates. Technology and Disability. 2013;25(3):167-76.

12. Gibson G, Newton L, Pritchard G, Finch T, Brittain K, Robinson L. The provision of assistive technology products and services for people with dementia in the United Kingdom. Dementia (London). 2016;15(4):681-701.

---

## [Decision Letter · Decision Letter 1]

7 Sep 2022

PONE-D-22-07145R1Informal caregivers and assistive technology in Norwegian nursing homes.PLOS ONE

Dear Dr. Anker-Hansen,

Thank you for submitting your manuscript to PLOS ONE. After careful consideration, we feel that it has merit but does not fully meet PLOS ONE’s publication criteria as it currently stands. Therefore, we invite you to submit a revised version of the manuscript that addresses the points raised during the review process.

 Both Reviewers have suggested further minor revision. I encourage Authors to include them in a revised version. 

We look forward to receiving your revised manuscript.

Kind regards,

Stefano Triberti, Ph.D.

Academic Editor

PLOS ONE

Journal Requirements:

Reviewers' comments:

Reviewer's Responses to Questions

**Comments to the Author**

1. If the authors have adequately addressed your comments raised in a previous round of review and you feel that this manuscript is now acceptable for publication, you may indicate that here to bypass the “Comments to the Author” section, enter your conflict of interest statement in the “Confidential to Editor” section, and submit your "Accept" recommendation.

Reviewer #1: (No Response)

Reviewer #2: (No Response)

2. Is the manuscript technically sound, and do the data support the conclusions?

Reviewer #1: Yes

Reviewer #2: Yes

3. Has the statistical analysis been performed appropriately and rigorously? 

Reviewer #1: N/A

Reviewer #2: Yes

4. Have the authors made all data underlying the findings in their manuscript fully available?

Reviewer #1: Yes

Reviewer #2: Yes

5. Is the manuscript presented in an intelligible fashion and written in standard English?

Reviewer #1: Yes

Reviewer #2: Yes

6. Review Comments to the Author

Reviewer #1: Thank you for the opportunity to rereview.

My comments have been addressed. I have a few minor revisions:

'them self' in the addition about Gibson in the DISCUSSION should be 'themself'.

Also 'That used on persons with dementia were devices and systems used by informal or formal caregiver to care for a person with dementia; examples are monitoring systems, environmental sensors, cameras, alarms and so on.'

I would suggest instead: Technologies used on persons with dementia were devices and systems used by informal or formal caregivers to care for a person with dementia; dementia; examples

are monitoring systems, environmental sensors, cameras and alarms.' (add 'Technologies' and 's' to caregiver, remove 'and so on' as these are given as examples so others would be expected).

Also in DISCUSSION: 'it was decided to use a coercive practice was used and locking residents inside their rooms.'

needs correcting to 'it was decided to use a coercive practice whereby residents were locked inside their rooms.'

or (I would prefer): 'it was decided to contain residents by locking them inside their rooms.'

In LIMITATIONS: 'First, the small sample size (11 participants) limits the findings to similar populations and contexts.'

I would put (11 participants in 2 care homes)

Also in LIMITATIONS I think 'did not have any direct experience of it' would be better.

Reviewer #2: I did not see the supplemental material which supposedly includes the interview schedule and follow-up questions. Authors should be sure that it is included and refer the reader to the supplement.

7. PLOS authors have the option to publish the peer review history of their article (what does this mean?). If published, this will include your full peer review and any attached files.

Reviewer #1: No

Reviewer #2: No

---

## [Author Response · Author response to Decision Letter 1]

8 Sep 2022

Comments Answers 

'them self' in the addition about Gibson in the DISCUSSION should be 'themself'.

Also 'That used on persons with dementia were devices and systems used by informal or formal caregiver to care for a person with dementia; examples are monitoring systems, environmental sensors, cameras, alarms and so on.'

I would suggest instead: Technologies used on persons with dementia were devices and systems used by informal or formal caregivers to care for a person with dementia; dementia; examples

are monitoring systems, environmental sensors, cameras and alarms.' (add 'Technologies' and 's' to caregiver, remove 'and so on' as these are given as examples so others would be expected). Amended in accordance with the suggestions

Also in DISCUSSION: 'it was decided to use a coercive practice was used and locking residents inside their rooms.'

needs correcting to 'it was decided to use a coercive practice whereby residents were locked inside their rooms.'

or (I would prefer): 'it was decided to contain residents by locking them inside their rooms. Changed to 'it was decided to contain residents by locking them inside their rooms.

In LIMITATIONS: 'First, the small sample size (11 participants) limits the findings to similar populations and contexts.'

I would put (11 participants in 2 care homes)

Also in LIMITATIONS I think 'did not have any direct experience of it' would be better. Amended according to the suggestions

Reviewer #2: I did not see the supplemental material which supposedly includes the interview schedule and follow-up questions. Authors should be sure that it is included and refer the reader to the supplement.

 The interview guide was uploaded during the submission process. The reader is refered to the supplement at suggested with the following sentence: see supplement for interview schedule

---

## [Editor Report · Decision Letter 2]

19 Sep 2022

Informal caregivers and assistive technology in Norwegian nursing homes.

PONE-D-22-07145R2

Dear Dr. Anker-Hansen,

We’re pleased to inform you that your manuscript has been judged scientifically suitable for publication and will be formally accepted for publication once it meets all outstanding technical requirements.

Kind regards,

Stefano Triberti, Ph.D.

Academic Editor

PLOS ONE
---

## [Editor Report · Acceptance letter]

26 Sep 2022

PONE-D-22-07145R2 

Informal caregivers and assistive technology in Norwegian nursing homes. 

Dear Dr. Anker-Hansen:

I'm pleased to inform you that your manuscript has been deemed suitable for publication in PLOS ONE. Congratulations! Your manuscript is now with our production department. 

Kind regards, 

on behalf of

Dr. Stefano Triberti 

Academic Editor

PLOS ONE